# Factor Structure and Measurement Invariance of the Very Short Form of Infant Behavior Questionnaire-Revised (IBQR-VSF): A Study among Vietnamese Children

**DOI:** 10.3390/healthcare10040689

**Published:** 2022-04-06

**Authors:** Mizuki Takegata, Yukiko Ohashi, Hien Anh Thi Nguyen, Michiko Toizumi, Hiroyuki Moriuchi, Duc Anh Dang, Lay-Myint Yoshida, Maria A. Gartstein, Samuel Putnam, Toshinori Kitamura

**Affiliations:** 1Department of Pediatric Infectious Diseases, Institute of Tropical Medicine, Nagasaki University, Nagasaki 852-8523, Japan; toizumi@nagasaki-u.ac.jp (M.T.); lmyoshi@nagasaki-u.ac.jp (L.-M.Y.); 2Kitamura Institute of Mental Health Tokyo, Tokyo 151-0063, Japan; y-ohashi@jiu.ac.jp (Y.O.); kitamura@institute-of-mental-health.jp (T.K.); 3Faculty of Nursing, Josai International University, Chiba 151-0063, Japan; 4National Institute of Hygiene and Epidemiology, Hanoi 100000, Vietnam; ntha@nihe.org.vn (H.A.T.N.); dangducanh.nihe@gmail.com (D.A.D.); 5Department of Pediatrics, Nagasaki University Graduate School of Biomedical Sciences, Nagasaki 852-8102, Japan; hiromori@nagasaki-u.ac.jp; 6Department of Psychology, College of Arts and Sciences, Washington State University, Pullman, WA 644820, USA; gartstma@wsu.edu; 7Department of Psychology, Bowdoin College, Brunswick, ME 04011, USA; sputnam@bowdoin.edu; 8Kitamura KOKORO Clinic Mental Health, Tokyo151-0063, Japan; 9Department of Psychiatry, Graduate School of Medicine, Nagoya University, Nagoya 466-8550, Japan; 10T. and F. Kitamura Foundation for Studies and Skill Advancement in Mental Health, Tokyo 151-0063, Japan

**Keywords:** factor structure, IBQ-R VSF, measurement invariances, Vietnamese

## Abstract

The Infant Behavior Questionnaire-Revised (IBQ-R) assesses the temperament of infants in Western and non-Western countries. Although its factor analyses revealed three factors—surgency, negative affectivity, and effortful control—in the Western culture, the degree to which these are universal or culturally specific is unclear. This study developed a Vietnamese version of the IBQ-Revised Very Short Form (R-VSF) and examined its factor structure in a Vietnamese population. The Vietnamese IBQ-R VSF was administered to 292 mothers of infants between the ages of 3 and 18 months in Nha Trang city, Vietnam, between July and September 2019. After deleting items to achieve sufficient Cronbach’s alphas for each scale (surgency, negative affectivity, and orienting/regulation), the remaining 28 items were aggregated to parcels subjected to exploratory factor analyses (EFAs). EFAs revealed a 3-factor model corresponding to the original theory, and confirmatory factor analyses indicated a good fit of this structural model. The final 3-factor model with parcels indicated measurement and structural invariance between mothers of boys and girls.

## 1. Introduction

Although infant temperament has a pivotal role in developmental psychology and child psychiatry research, its definition has long been debated [1]. There is a degree of consensus among contemporary researchers that temperament refers to early appearing and relatively stable individual differences in the domains of activity, attention, emotional expression, and self-regulation; with these dispositions developing as the product of complex interactions among genetic, biological, and environmental factors [2]. The presumption of biological bases seems to connote that the underpinnings of temperament are universal across different cultures. The current study explores this notion, examining the replicability of temperament structure in a culture distinct from that in which it was first identified.

The Infant Behavior Questionnaire (IBQ) [3] was developed to assess the temperament of infants between the ages of 3 and 12 months and included subscales measuring activity level, smiling and laughter, fear, distress to limitations, duration of orienting, and soothability. The instrument was later modified to create the IBQ-Revised (IBQ-R) [4], with the addition of subscales assessing approach, vocal reactivity, perceptual sensitivity, sadness, falling reactivity, and cuddliness and with independent scales for pleasure in high- and low-intensity situations. Factor analyses of these subscales revealed three broad subscales of surgency (approach-based tendencies in activity and positive affect in intense contexts), negative affectivity (frequent expressions of distress), and orienting/regulation (indicators of attention and enjoyment of low-intensity stimuli). Later, the IBQ-R was further modified, eliminating items to produce Short and Very Short Forms [5]. The Very Short IBQ-R (IBQR-VSF) utilized in the current study consists of 37 items that form three scales representing surgency, negative affectivity, and orienting/regulation. The IBQ, IBQ-R, and IBQ-R VSF have subsequently demonstrated good reliability and validity [6,7,8,9].

Although the IBQ-R VSF has been translated into various languages and used in several countries, consensus has not been reached regarding its psychometric characteristics. The measurement invariance of the 3-factor structure of the IBQR-VSF was reported by Leekes et al. (2017) [10], among an American population, whereas Peterson et al. (2017a, 2017b) [11,12] in New Zealand suggested that the 5-factor model fit the data better and revealed invariance of the factor structure across European, Maori, Pacific, and Asian populations. Nevertheless, the goodness-of-fit of the Peterson et al. 5-factor model (CFI = 0.77) was far below usually expected values (0.95 proposed by Hu and Bentler (1999) [13] as a cut-off point).

There may be several reasons for this low goodness-of-fit estimate. First, a large number of indicators (items) may distort the item number/factor ratio. This may produce substantial covariance of error terms, violating the proposition of structural equation modeling. Second, a 7-point scale used for each IBQR-VSF item and extreme skewness of some items may violate the normality of the item distribution. To remedy these drawbacks, some authors have proposed creating parcels. Parceling is an aggregation of individual items into one or more composite variables [14]. This approach results in a lower item/factor ratio, greater normality of item parcels, and lesser covariance between error terms. Confirmatory factor analyses (CFAs) using item parcels often provides more accurate estimation of measurement structure compared with those using the items. Hence, parceling has been increasingly used in studies of psychometric properties.

Of additional importance is whether the manifestation of infant temperament in Asian cultural backgrounds is the same as in Western societies. Although the temperament of children and its structures maybe universal across different cultures, behavioral expression is likely to vary among different cultural backgrounds. Studies to designed to confirm (or refute) the factorial structure of a psychological measure may be complemented with an initial exploration of scale items to identify those that represent the concept of interest and elimination of items that appear idiosyncratic to specific cultural (and linguistic) backgrounds.

Our present study aimed to develop the Vietnamese version of the IBQ-R VSF and examine its factor structure in a Vietnamese population, first eliminating items detracting from the measurement of the intended constructs in this unique culture. The research question of this study was whether the Vietnamese version has a 3-factor structure that is consistent with the original IBQ-R VSF.

## 2. Materials and Methods

### 2.1. Study Procedures and Participants

The validation study was conducted as part of a larger historical cohort study investigating the relationship between hypertensive disorders of pregnant mothers and infant development in Nha Trang city, Viet Nam, between July and September 2019. Participants of this validation study were mothers of infants/toddlers aged 3 to 18 months born in Khanh Hoa General Hospital. First, demographic information of infants/toddlers and their mothers, including pregnancy outcome and obstetric information, were obtained from hospital data. Second, candidates were recruited by eight research assistants via telephone calls. If the mothers agreed and gave informed consent, the research assistants visited their homes to interview them using a set of questionnaires.

### 2.2. Measurements

The original English version of IBQR-VSF [5] was translated into Vietnamese by two independent bilingual translators and back-translated by two other translators (who were unaware of the original English version). The original author confirmed this final version. Five Vietnamese members of the Cooperative Office of Khanh Hoa Health Service confirmed the comprehensiveness of all items of the Vietnamese IBQR-VSF.

### 2.3. Data Analysis

First, we calculated the mean, standard deviation, skewness, and kurtosis of all the IBQR-VSF items. Since we aimed to select sets of Vietnamese IBQR-VSF items representing each subscale with a single-factor structure, we computed Cronbach’s alpha coefficient of the items of IBQR-VSF subscales separately for surgency, negative affect, and orienting/regulation [5]. Next, we eliminated items one after the other to check whether doing so increased Cronbach’s alpha coefficient, until no additional deletions would improve the alpha coefficient.

Then we created three item parcels using a factor algorithm [14] for each subscale and performed EFA with these item parcels. The factor of the whole data (*N* = 292) was examined using the Kaiser–Meyer–Olkin (KMO) index and Bartlett’s sphericity test [15]. Factor extraction was performed using the maximum-likelihood method with PROMAX rotation. The EFA-derived models were compared using a series of CFAs. The 1-factor model was most parsimonious; the 2-factor model was acceptable for the model fit only when χ^2^ reduction per *df* change was statistically significant. This procedure was repeated until no statistically significant decrease in χ^2^ was observed.

After deciding the best-fit model, we examined the model’s configural, factor, and structural invariances across mothers’ reports of boys and girls. We started from the configural invariance through metric, scalar, residual, and factor variance invariances to factor covariance invariances [16]. Configural invariance means that boys and girls showed the same indicators and factors. Metric invariance means that the factor loadings of the like indicators are not variant between boys and girls. Scalar invariance means that intercepts of like indicators are not variant between boys and girls. Residual invariance means that residuals (errors) of like indicators are not variant between boys and girls. Factor variance invariance means that variances of like factors are invariant between boys and girls. Factor covariance invariance means that factor covariances of like factors are not variant between boys and girls [16]. We divided the participants into two groups of sex because child temperament is different between boys and girl [17]. According to the previous study, girls showed higher levels of sociability and lower levels of overall negative emotionality, sadness, anger, and impulsivity than boys [17]. The progress from one step to the next was accepted if (a) the χ^2^ decrease was not significant for the *df* difference, (b) decrease in CFI was less than 0.01, or (c) increase in root mean square error of approximation (RMSEA) was less than 0.015 [18]. Notably, a χ^2^ reduction is known to be sensitive to the sample size (*N*) and produces an unnecessary rejection of invariance.

## 3. Results

The items of the IBQR-VSF were filled in by 292 mothers of children who were between 3 and 18 months of age. Out of 292 mothers, most of them were married (*N* = 291, 99.7%), with average age of 28.57 (standard deviation = 5.01). The number of male children was 110 (37.7%), while that of female children was 112 (38.4%). While all of them filled in 20 or more items, 278 (62%) mothers filled in 30 or more items. The number of mothers who had a female child was 112 (40.28%), and those of male child was 111(39.92%), out of 287 participants, while 55 failed to answer the item. For EFAs, Little’s missing completely at random (MCAR) test was *p* < 0.001, and therefore missing values were not MCAR, so missing values were not imputed. For CFAs, AMOS provided full information likelihood estimation.

Of the IBQR-VSF items, four and five items in surgency and orienting/regulation showed skewness > 2.0, respectively; however, these items were not excluded or transformed for normality. None of the negative affect items showed extraordinarily high skewness (Table 1).

Cronbach’s alpha coefficients were initially 0.407, 0.743, and 0.299 for surgency, negative affect, and orienting/regulation, respectively (Table 2). Repeated deletion of items one after the other increased Cronbach’s alpha coefficients to 0.769, 0.770, and 0.596 for surgency (items 7, 15, 20, and 27 were deleted), negative affect (items 3, 17, and 23 were deleted), and orienting/regulation (item 11 was deleted), respectively (Appendix A).

We created three, three, and two item parcels (factor algorithm) for surgency (S1–S3), negative affect (N1–N3), and orienting/regulation (OR1–OR3), respectively. These item parcels were entered into the EFAs (Table 3). We calculated a 4-factor EFA but its fourth factor failed to have more than one indicator with a factor loading > 0.33.

In CFAs of item parcels, fit with the data was improved significantly (*p* < 0.001) from the 1-factor (χ^2^ = 251.086, *df* = 27) to the 2-factor (χ^2^ = 95.056, *df* = 26) models. It was also significantly (*p* < 0.001) improved from the 2-factor to the 3-factor models (χ^2^ = 37.571, *df* = 24). The 3-factor model fit well: χ^2^/*df* = 1.56, CFI = 0.970, and RMSEA = 0.044. This showed that the parcels loaded high on the theoretically expected factors (Table 3). There appeared slight to moderate correlations between the factors: S vs. N = 0.20, N vs. OR = 0.27, and S vs. OR 0.49. (Figure 1).

The final 3-factor model indicated invariance between mothers with boys and girls up to factor covariance invariance (Table 4). When comparing the factor means between boys and girls, there were no significant differences between the two sexes.

## 4. Discussion

The present study demonstrated that the temperament of Vietnamese children could be effectively measured using the three temperament domains presented by Rothbart and colleagues [5,6]. The 3-factor structure among the Vietnamese children showed configural, measurement, and structural invariances between boys and girls. Our findings complement those of Leerkes et al. (2017), who examined multi-group confirmatory factor analysis comparing European American and African American mothers, demonstrating the 3-factor structure of the IBQR (positive affectivity/surgency, negative affectivity, orienting/regulatory capacity) across mothers from these different backgrounds. Our study also supports the finding that the IBQR-VSF is appropriate for use among the Vietnamese population.

However, there appeared to be cultural differences in the meanings of some items. The missing rate was different among the IBQR-VSF items. For instance, it was very high (66%) for item 5 (“How often during the last week did the baby enjoy being read to?”). Although this item may reflect effortful control in certain cultures, the manifestation of effortful control through this behavior could be dependent on the cultural context. Despite the recent high literacy rate, reading a book to a baby is not a common practice in Vietnam. Among the original IBQR-VSF items, some items reduced internal consistency, particularly those connected with rearing customs (e.g., item 23, infant seat, playpen, car seat, etc.) and lifestyle factors (e.g., item 15, ringing of telephone; item 27, airplane passing overhead). For example, car is becoming common, but most Vietnamese still use motor-bikes (item 23). Another example is that people may not be familiar with airplanes in such a place where there is no airport. Hence, these items may sound unfamiliar to a Vietnamese population due to the different lifestyles (See Appendix A).

Limitations of the current study should be noted. First, we were not able to obtain the exact number of recruited participants; hence, the demographic characteristics may be slightly different from the total population in Nha Trang city. Second, we relied only on parental reports as a measure of child temperament. Direct observation of children and its association with parental reports should be investigated in future studies. Third, we used only mothers as informants of the children. Fathers may provide a different perspective (Kitamura et al. 2015) [19]. Children may be viewed differently by fathers and mothers. Alternatively, children may behave differently in front of fathers and mothers. Fourth, the results of the IBQR-VSF should have been compared with other temperament measures (concurrent validity). Fifth, we dealt with the scores of the instrument as being dimensional. Hence, the primary purpose of this study was to confirm the 3-factor structure of temperament of children. Temperament of children may also be viewed in terms of categories [20], providing a different perspective.

Despite these drawbacks, our study demonstrated that the 3-factor structure was observed among Vietnamese children using the Vietnamese version of the IBQR-VSF.

## 5. Conclusions

The present study suggested that the structure of the IBQR-VSF among Vietnamese children corresponded to the original 3-factor model found among the US population. However, their manifestation differed somewhat, requiring that some items from the original item pool be eliminated. Overall, this abbreviated version might be recommended to measure the three child temperament domains in the same framework that is used in the US culture. Additionally, this Vietnamese version can be used for health professionals/research experts to assess child’ temperament in order to provide support/advice on families as well to assess effects of intervention related to child development.

## Figures and Tables

**Figure 1 healthcare-10-00689-f001:**
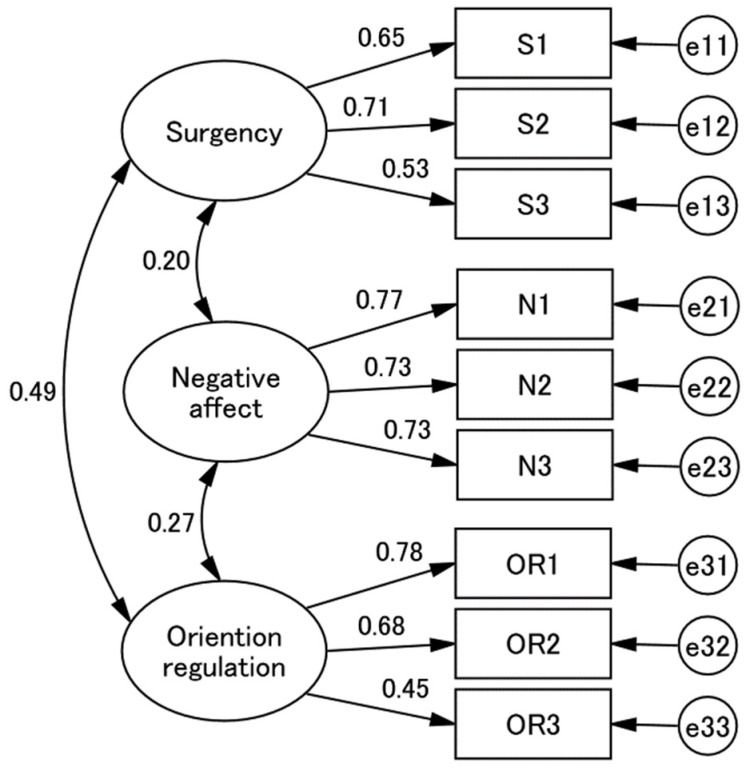
CFA with 3-factor model with parcels. The three factors are surgency (S), negative affect (N), and Orientation regulation (O). Parcels were created by random algorithms; S1 = item 1 and 13, S2 = item 26 and 37, N1 = item 4 and 9, N2 = item 10 and 16, N3 = item 29 and 33, e1 = item 24 and 25, e2 = item 30 and 34. χ^2^, *df*, comparative fit index (CFI) and root mean square error appropriation (RMSEA) were stated as model fit indicators.

**Table 1 healthcare-10-00689-t001:** Mean, SD, skewness, and kurtosis of IBQR-VSF items (*N* = 292).

No. of Item	Item	*N*	Mean	SD	Skewness	Kurtosis
Surgency						
1	The baby squirmed and/or tried to roll away.	291	4.75	2.50	0.5	−1.5
2	The baby laughed when being tossed around playfully.	240	6.79	0.69	4.8	26.6
7	The baby moved quickly toward new objects.	246	5.75	1.86	1.5	1.0
8	The baby laughed put into the bath water.	292	6.45	1.37	3.1	8.8
13	The baby squirmed and/or turned body when placed on his/her back.	292	5.47	2.21	1.2	−0.3
14	The baby squirmed during a peekaboo game.	288	6.80	0.62	5.4	39.8
15	The baby looked up from playing when the telephone rings.	277	6.38	1.33	2.8	7.5
20	The baby got excited about exploring new surroundings.	280	5.53	1.70	1.0	−0.1
21	The baby smiled or laughed when given a toy.	287	5.82	1.59	1.8	2.5
26	The baby vocalized when hair was washed.	287	5.02	2.41	0.7	−1.2
27	The baby noticed the sound of an airplane.	194	5.23	1.87	1.0	0.0
36	The baby made talking sounds when riding in a car.	284	5.01	2.15	0.8	−0.8
37	The baby squirmed and turned body when placed in an infant seat or car seat.	274	4.48	2.17	0.5	−1.3
	Negative affect					
3	The baby showed distress when tired.	269	3.47	2.13	0.4	−1.2
4	The baby clung to a parent when introduced to an unfamiliar adult.	277	3.23	2.27	0.6	−1.3
9	The baby whimpered or sobbed when it was time for bed or a nap.	282	3.52	2.20	0.4	−1.4
10	The baby cried if someone does not come within a few minutes.	288	3.57	2.29	0.4	−1.5
16	The baby got angry (crying and fussing) when left in the crib?	289	3.90	2.17	0.2	−1.4
17	The baby startled at a sudden change in body position.	282	2.55	1.70	0.9	−0.3
22	The baby became tearful at the end of an exciting day.	289	2.10	1.57	1.6	1.8
23	The baby protested being placed in a confining place.	272	4.27	2.08	0.3	−1.3
28	The baby refused to go to the unfamiliar person.	268	3.93	2.16	0.2	−1.4
29	The baby cried when parents were busy.	283	4.17	2.18	0.0	−1.6
32	The became upset when s/he could not get what s/he wanted.	281	4.90	1.96	0.6	−0.8
33	The baby clung to a parent in the presence of unfamiliar adults.	281	3.21	2.16	0.5	−1.2
	Orienting/regulation					
5	The baby enjoyed being read to.	98	5.79	1.85	1.5	1.1
6	The baby played the baby play with one toy or object.	282	4.55	2.18	0.4	−1.3
11	The baby seemed eager to get away as soon as the feeding was over.	242	3.88	2.33	0.1	−1.5
12	The baby soothed immediately when singing to him/her.	291	6.19	1.09	2.3	6.5
18	The baby enjoyed hearing the sound of words.	256	6.16	1.19	2.3	6.0
19	The baby looked at pictures in books and/or magazines.	243	4.23	2.33	0.3	−1.6
24	The baby seemed to enjoy him/herself When being held.	291	5.78	1.48	1.2	0.8
25	The baby soothed when showing the baby something to look at.	288	6.26	0.99	2.2	6.7
30	The baby enjoyed gentle rhythmic activities, such as rocking or swaying.	289	6.32	0.99	2.6	9.3
31	The baby stared at a mobile, crib bumper or picture.	282	4.72	2.28	−0.7	−1.1
34	The baby seemed to enjoy him/herself when rocked or hugged.	292	6.03	1.30	−1.9	3.9
35	The baby soothed when patting or gently rubbing some part of the baby’s body.	291	6.22	1.11	−2.4	6.9

SD = standard deviation; *N* = number.

**Table 2 healthcare-10-00689-t002:** Items deleted until Cronbach’s α value increased.

Items	α
Surgency	
IB1, IB2, IB7, IB8, IB13, IB14, IB15, IB20, IB21, IB26, IB27, IB36, IB37	0.407
IB1, IB2, IB7, IB8, IB13, IB14, IB15, IB21, IB26, IB27, IB36, IB37	0.440
IB1, IB2, IB8, IB13, IB14, IB15, IB21, IB26, IB27, IB36, IB37	0.512
IB1, IB2, IB8, IB13, IB14, IB21, IB26, IB27, IB36, IB37	0.543
IB1, IB2, IB8, IB13, IB14, IB21, IB26, IB36, IB37	0.586
Negative affect	
IB3, IB4, IB9, IB10, IB16, IB17, IB22, IB23, IB28, IB29, IB32, IB33	0.743
IB4, IB9, IB10, IB16, IB17, IB22, IB23, IB28, IB29, IB32, IB33	0.754
IB4, IB9, IB10, IB16, IB17, IB22, IB28, IB29, IB32, IB33	0.769
IB4, IB9, IB10, IB16, IB22, IB28, IB29, IB32, IB33	0.770
Orienting regulation	
IB6, IB11, IB12, IB18, IB19, IB24, IB25, IB30, IB31, IB34, IB35	0.299
IB6, IB12, IB18, IB19, IB24, IB25, IB30, IB31, IB34, IB35	0.596

**Table 3 healthcare-10-00689-t003:** Exploratory factor analyses of the IBQR-VSF.

	1 Factor	2 Factors		3 Factors		
	I	I	II	I	II	III
S1	−0.02	−0.12	**0.67**	−0.02	**0.79**	−0.10
S2	0.14	0.06	**0.68**	0.06	**0.57**	0.20
S3	0.03	−0.05	**0.60**	0.03	**0.65**	−0.05
N1	**0.69**	**0.69**	−0.02	**0.72**	0.03	−0.06
N2	**0.82**	**0.84**	−0.05	**0.81**	−0.05	0.05
N3	**0.79**	**0.78**	0.04	**0.79**	0.07	−0.00
OR1	0.12	0.06	**0.50**	−0.08	0.27	**0.52**
OR2	0.19	0.14	**0.51**	−0.02	0.24	**0.57**
OR3	0.29	0.28	0.11	0.04	−0.29	**0.75**

S (Surgency)1 = IB1 + IB26 + IB21, S2 = IB13 + IB36 + IB8, S3 = IB37 + IB2 + IB14. N (Negative Affect)1 = IB33 + IB10 + IB22, N 2 = IB4 + IB16 + IB9, N3 = IB29 + IB32 + IB28. OR (Orienting/Regulation)1 = IB31 + IB35 + IB30, OR2 = IB19 + IB18 + IB25, OR3 = IB6 + IB24 + IB34 + IB12. Factor loadings with 0.3 or more are in bold.

**Table 4 healthcare-10-00689-t004:** Configural, measurement, and structural invariances between boys and girls.

Models	χ^2^	*df*	χ^2^/*df*	χ^2^ (*df*)	CFI	ΔCFI	RMSEA	ΔRMSEA	
Configural	53.285	48	1.110	Ref	0.984	Ref	0.022	Ref	Accept
Metric	69.535	54	1.288	16.250 (6) *	0.953	0.041	0.036	0.004	Accept
Scalar	83.831	63	1.331	14.296 (9) NS	0.937	0.015	0.039	0.003	Accept
Residual	99.985	72	1.389	16.154 (9) NS	0.915	0.022	0.042	0.003	Accept
Factor variance	104.371	75	1.392	4.386 (3) NS	0.911	0.004	0.042	0.000	Accept
Factor covariance	104.968	78	1.346	0.597 (3) NS	0.918	+0.007	0.040	+0.002	Accept

* *p* < 0.05.

## Data Availability

Due to the nature of this research, participants of this study did not agree for their data to be shared publicly, so supporting data are not available.

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
