# Peer review of "Factor Structure and Measurement Invariance of the Very Short Form of Infant Behavior Questionnaire-Revised (IBQR-VSF): A Study among Vietnamese Children"

_healthcare, 2022, doi:10.3390/healthcare10040689_

Round 1
Reviewer 1 Report
In general, I find this article to be well written. This is a well-prepared study, with a solid statistical approach. I do find this paper to be a good discussion issue. However, the paper presents some weaknesses, and I would suggest some minor revisions:
Methods. The description of the participants is very general. Although the abstract indicates that 292 mothers participated, page 3 does not even show the number of mothers who participated. The authors must provide, either in the text or in a table, the sociodemographic characteristics of the sample that participated in this study.
Conclusions. The authors should expand the conclusions indicating the usefulness and implications of their study for practice, both in the evaluation of child behavior and in relation to the development of intervention/advice plans for families.
Formal aspects. On page 7, the authors refer to appendix 1, but it does not appear in the article or in the supplementary material. On page 8, the appendix is listed as A.
I would like to thank the Editor for the opportunity to review this study and I am flattered to be able to provide my contribution.
Author Response
Response to Reviewer’ 1 comments.
Thank you so much for your valuable comments. We corrected the manuscript according to your comments.
- The description of the participants is very general. Although the abstract indicates that 292 mothers participated, page 3 does not even show the number of mothers who participated. The authors must provide, either in the text or in a table, the sociodemographic characteristics of the sample that participated in this study.
Thank you. We added some sociodemographic characteristics in the result section accordingly (not the method section).
2.Conclusions. The authors should expand the conclusions indicating the usefulness and implications of their study for practice, both in the evaluation of child behavior and in relation to the development of intervention/advice plans for families.
Thank you so much. Following was added in conclusion part.
Also, this Vietnamese version can be used for health professionals/research experts to assess child’ temperament in order to provide support/advice on families as well to assess effects of intervention related to child development.
- Formal aspects. On page 7, the authors refer to appendix 1, but it does not appear in the article or in the supplementary material. On page 8, the appendix is listed as A.
Thank you again.

Reviewer 2 Report
Good research, a little bit of information on the background of the sample/recruiting (who is visiting the hospital? which problems? How many (%) of those seen agreed to participate..
Author Response
Response to Reviewer’ 2 comments.
Thank you so much for your valuable comments. We corrected the manuscript according to your comments.
- Good research, a little bit of information on the background of the sample/recruiting (who is visiting the hospital? which problems? How many (%) of those seen agreed to participate..
Thank you for your valuable comments. Unfortunately, we could not get the exact number actually recruited since the research assistants collaborated with staff in community health centers. We acknowledge that it is a limitation. Although the exact data is not available, the number of rejections was low according to our research assistants. Following sentence was added in Limitation part;
First, we were not able to obtain the exact number of recruited participants, hence, the demographic characteristics may be slightly different from the total population in Nha Trang city.
Thank you again.
